# Gay Sex Workers in China’s Medical Care System: The Queer Body with Necropolitics and Stigma

**DOI:** 10.3390/ijerph17218188

**Published:** 2020-11-05

**Authors:** Eileen Yuk-ha Tsang

**Affiliations:** Department of Social and Behavioral Sciences, City University of Hong Kong, Tat Chee Avenue, Kowloon Tong, Hong Kong; eileen@cityu.edu.hk

**Keywords:** necropolitics, stigma, gay sex workers, *hukou*, medical care, China, HIV

## Abstract

The struggles of China’s gay sex workers—men who sell sex to other men—illustrate how the multi-layered stigma that they experience acts as a form of necropolitical power and an instrument of the state’s discrimination against gay sex workers who are living with HIV. One unintended side effect of this state power is the subsequent reluctance by medical professionals to care for gay sex workers who are living with HIV, and discrimination from Chinese government officers. Data obtained from 28 gay sex workers who are living with HIV provide evidence that the necropower of stigma is routinely exercised upon the bodies of gay sex workers. This article examines how the necropolitics of social death and state-sanctioned stigma are manifested throughout China’s health system, discouraging gay sex workers from receiving health care. This process uses biopolitical surveillance measures as most of gay sex workers come from rural China and do not enjoy urban *hukou*, thus are excluded from the medical health care system in urban China. Public health priorities demand that the cultured scripts of gendered Chinese citizenship must reevaluate the marking of the body of gay sex workers as a non-entity, a non-human and socially “dead body.”

## 1. Introduction

In 2011, China’s Premier Wen Jiaobo visited the Centers for Disease Control and Prevention (CDC) in Beijing and vowed to provide funding and policy support to research drugs and vaccines guaranteed to improve care for HIV patients. He confirmed that the Chinese government would provide more funding and strong policy support to guarantee improvements in care for patients and research into drugs and vaccines. He also promised to fight poverty in areas where HIV was prevalent, to provide stronger societal support for AIDS prevention, and to fully implement the Four Free One Care Policy [1]. The Four Free One Care Policy means free treatment, free voluntary counseling and testing (VCT), free prevention of mother–child transmission (PMCT), free schooling for AIDS orphans, and provision of social relief for HIV patients [1]. According to the Chinese Constitution Article 55, medical doctors refusing to provide treatment or conduct surgery on someone who is living with HIV are subject to penalties such as medical license disqualification and even jail time [2]. In addition, Constitution No. 41 states that designated hospitals should provide counselling, diagnosis and treatment services. Hospitals are not allowed to refuse treatment for people who are living with HIV or AIDS [2]. Nevertheless, stigma against gay males, lesbians, bisexuals, transgender people, and those living with HIV remains prevalent, perhaps aided by the ambiguous wording of the law.

However, the reality is that gay sex workers (GSWs) who are living with HIV face different realities than that which are stipulated by China law. Round and Kuznetsova [3] argue that untold numbers of migrants who enter spaces illegally are subjected to discrimination and social death, as they are ushered into conditions that deny them autonomy. Social death connotes that, although the GSWs are still living, they face discrimination in various forms. Anecdotally, GSWs with HIV say that hospital doctors and staff noticeably avoid initiating conversation, asking questions, or even making eye contact with them. Likewise, government officers and even police often do not actively try to help or assist. Most troubling is that family members and even the gay community do not accept and acknowledge them. In China, migrants have limited autonomy because the *hukou* system (household registration system) denies them free access to social services anywhere except the town or province where they are registered. The *hukou* is the household registration system which provides a permanent identification number, comparable to a birth certificate and social security number in the United States or the permanent identification card in England. The rural *hukou* means that urban medical services charge them full price for medications and treatments which the rural MSWs typically cannot afford. This situation reflects larger social forces referred to as biopolitical measures of control. These controls are exercised with respect to the GSWs living with HIV who are consequently constructed as diseased and criminal, and forced to go to their hometowns for treatment.

A balanced application of Mbembe’s [4] necropolitics includes an explanation of the mechanisms used for resistance and agency. Many scholars [5,6] note that sex workers—gay and straight—often do not see themselves as simply waiting for physical death, but as workers with an agency in charge of their lives. Therefore, necropolitics is a useful anchor because it can accommodate examples of life-affirming agency and resistance. The marginalized communities of GSWs living with HIV face significant hardship, but they find ways to survive in the state-sanctioned medical care system.

Davies, Isakjee and Dhesi [7] argue that stigma-based political inaction towards migrants results in death. This death is caused by the institutional polices, medical care, and the attitudes of the government. In China, GSWs who migrate to the city for work are subject to conditions that prevent life from flourishing. The GSWs survive in their ‘underground’ trade by paying bribes because sex work is illegal in China. In bringing the queer to the necropolitical, Haritaworn, Knuntsman and Pocosso [8] argued that the ‘queer’ community is generally ignored by the government, particularly those who are patients living with HIV and trying to access health care. When urban GSWs with a rural *hukou* return to their hometown health facility, they must disclose details of their HIV-positive status, which has ramifications for their family and the local village. There is much emphasis on multi-layered stigma and how ignorance about lesbian, gay and transgendered patients on the part of medical professionals raises mortality rates [9,10,11,12].

This perspective helps fill the research gap and extends the emerging literature on HIV-infected GSWs by differentiating how *social death* is not *living death* and is influenced by felt and enacted stigmas in three ways: (1) facing both enacted and felt stigmas, (2) lacking an urban *hukou*, and (3) being marginalized citizens under China’s medical care system.

### 1.1. Holistic Approach: The Necropolitics of Social Death

#### 1.1.1. Mbembe’s Theory of Necropolitics

The literature on necropolitics has mostly focused on Western countries. “Necropolitics” refers to the predicament of underprivileged groups excluded from the mainstream and waiting for death. This death typically refers to *physical death*, as certain groups are typically denied legal, biomedical, cultural recognition and basic human rights [4,7,13,14,15]. Made popular by Achille Mbembe [4], the theory of necropolitics incorporates the Foucauldian notion of biopower, which divides populations into conditions of life and death [4]. Even in liberal democracies, the state manages people and—either overtly or covertly—earmarks certain bodies for death. “Biopolitics” refers to the management and regulation of lifeworlds as a mechanism of state power and reveals the power dynamic between the ruling (powerful) groups and the ruled (underprivileged). Mbembe offered that necropolitics accounts for new and unique forms of social existence, “in which vast populations are subjected to conditions of life conferring upon them the status of the living dead.” Marginalized groups include migrants, those infected with HIV, sex workers, drug users, and LGBT groups [4].

#### 1.1.2. Necropolitics in Western Countries

Following Mbembe [4], Puar [16] argues that queerness is not about being lesbian or gay, but about pursuing activities that go against the state’s ambitions and norms. Coining the term “queer necropolitics,” Puar argues that non-heteronormative bodies are marked for death to support the state’s hegemony and nationalism. Debrix suggests that the horror of exposing people to death—the capacity to “let die”—leads to conditions in which human bodies are “unrecognizable, unidentifiable and sometimes indistinguishable from non-human matter” [17]. Both Puar and Debrix agree that the state’s abandonment of underprivileged groups pushes them to social and living death [8]. Queer underprivileged groups in Western countries experience stigma and discrimination from the government, their families, and their peers. In the field of queer necropolitics, McKinnon proposes that the mechanisms resulting in the death of those marked as queer can inadvertently result in conditions conducive to life. There is always “a life after death that the state cannot anticipate and does not wish” [13].

#### 1.1.3. The Necropolitics of Social Death in the Context of Stigma

Necropolitics helps explain the impact of stigma within the context of GSWs in today’s China. Stigma is generally defined as a social process of “othering, blaming, and shaming” that leads to status loss and discrimination [18,19]. Stigmatized groups are devalued through the exercise of social, cultural, economic, and political power [20]. Spurred from a socio-medical perspective, Scambler situated stigma within socio-cultural structural contexts and the interaction between stigma, class, capital and power. He noted the important difference between “felt stigma” from “enacted stigma.” *Felt stigma* (internal stigma or self-stigmatization) is the “fear of encountering discrimination and an internalized sense of shame” on the grounds of “socially unacceptable difference” [21]. It refers to the expectation of discrimination which prevents people from talking about their experiences or proactively seeking help. *Enacted stigma* (actual discrimination and unacceptability) refers to the experience of unfair treatment by others. Enacted stigma, as “structural or institutional discrimination,” is embedded in social, economic and political power [22]. Both felt stigma and enacted stigma are damaging when they result in withdrawal and restriction of social support. Smit et al. Smit, et al. [23] found that HIV-positive gay men who experienced higher levels of stigma were more likely to use illicit drugs. Where the prescription of PrEP is concerned, Goparaju et al. [24] notes that stigma from family members and the medical profession in the USA discourages women who are HIV negative from taking PrEP. It is important to consider the necropolitical power of fear which comes from discrimination (felt stigma).

In the case of the GSWs living with HIV, their homosexuality makes their bodies vessels of HIV infection which subjugates them to a “social death.” Both felt and enacted stigmas restrict their access to health care, facilitating necropolitical outcomes of being a positive carrier of HIV and destined to die. This study begins by acknowledging that the death worlds of GSWs who are living with HIV—which propagate both enacted and felt stigma—cannot be separated from the spatial codes that create death conditions.

#### 1.1.4. The Necropolitics of *Hukou*

In China, this global pattern of gendered sex work migration is manifested in a rural–urban *hukou* (household registration system) divide. The *hukou* is a key factor encouraging the marginalization and rural/urban segregation faced by the GSWs [25]. In the city, having a rural *hukou* can lead to alienation because it marks one as being a villager, and colloquially, a country bumpkin. Male migrant workers in urban China are typically underpaid [12] and afraid of their precarious access to social services locally [26]. In addition to institutionalized exclusion, the second-class citizenship of the GSWs is endorsed by the state’s neoliberal development discourses. Rural GSWs are often labelled as “low-suzhi (quality)”, “uncouth,” “provincial,” and their existence pollutes the social image in the city, encourages marginalization among those peasant male migrants in China.

In peasant migrant workers’ struggles against the lack of social security, Shi [26] argues that “workers’ bodies created a site for claim-making within the disciplinary space and had the potential of redeploying that space for their own ends...the workers had to subvert the physical boundaries that inscribed disciplinary power over their bodies”. The case of China’s rural GSWs is an even worse predicament than that of the factory workers. The necropolis created by felt and enacted stigmas, drug use and health abandonment clearly resist the spaces of urban homonormativity and heteronormativity in China.

In his ethnography of GSWs in urban China who lack urban *hukou*, Kong [25] notes that GSWs are often seen as “failed” gay citizens: “bad” gays that mix sex with money, corrupting the image of middle-class gay men. Scholars think *hukou* [11,12,25], cadres (government officers) and others—in system and out system [27]—are the three major boundaries to analyze social stratification in China. Scholars must address the gap in the international literature on the biopolitical and necropolitical conditions faced each year by hundreds—perhaps thousands—of homosexual rural-to-urban migrants in China.

### 1.2. The Legal Context of Sex Work and HIV

There are an estimated 20 million men having sex with men (MSM), an estimated 8 million GSWs in China, and many are rural-to-urban migrants away from their hometown in 2018 [28]. China’s Centre for Disease Control [29] estimated that more than 10% of the 8 million GSWs were living with HIV, approximately 850,000. In China, GSWs who migrate to the city for work are subject to conditions that prevent them from flourishing and being successful. The GSWs survive in their ‘underground’ trade by paying bribes because they are not entitled to *hukou* (household registration system) in the city [30], a form of biopolitical control of the rural poor. There is much emphasis on multi-layered stigma and how ignorance about lesbian, gay and transgendered patients on the part of medical professionals raises mortality rates [9,10,11,12]. Stigma and discrimination are expressed indirectly through institutional and social policies such as the *hukou* system which requires gay and transgendered patients to return to their hometown for treatments they cannot afford in the city. Stigma and discrimination are directly experienced in the interactions with medical personnel who make it clear that they disapprove of the GSWs lifestyle and even fear being accidentally infected.

## 2. Materials and Methods

Empirical data were obtained from 28 HIV-infected rural-to-urban GSWs. The interviewees were part of a larger study involving GSWs in northern China, and these 28 were a subset of a larger sample totaling 101. The 28 GSWs are biologically male and culturally identify themselves as homosexual men (gay), who prefer to have sex with other men. Of the 28, 15 were working in a high-end bar, and the other 13 were working in either a mid-tier or a low-end bar or worked freelance on the street. A four-year ethnographic study was conducted from May 2015 to August 2019. The researcher was introduced via a personal connection to the field site, a gay bar in Tianjin which was first visited in May 2015. The bar is popular and well known, located in a downtown area surrounded by several other clubs. The club is opulent and clean, designed to appeal to clientele who are relatively well off. As such, the club décor is like a casino or luxury hotel. Beginning at midnight, performers do their ‘shows.’ These can be drag shows but also striptease, simulated sex, and even Chinese opera. The club has a regular group of clients but also brings in new customers through advertising and word of mouth. The owner has a relationship with local police and is rarely raided. Given the illegal nature of sex work, individual client service activities occur in private rooms on the premises.

From May 2015 to Dec 2016, the researcher worked as an unpaid bartender in the club and given access to meet, interact, and interview 151 GSWs in the club. As part of the research, 20 of the 151 GSWs disclosed that they were living with HIV and accepted medication regularly, of which 15 agreed to subsequent interviews. A second round of data collection occurred from July 2017 to August 2019. A contact in Hong Kong connected the researcher to the 12 directors of other Tianjin-area gay NGOs. Follow-up arrangements were made to interview each of the 12 directors. With help from the 12 NGOs, the researcher met the GSWs in saunas, private clubhouses, public parks, on the street, and via the internet to diversify the samples. In total, the researcher talked to 158 GSWs in the second stage of data collection. Among the 158, 23 GSWs disclosed that they were living with HIV. They did not feel uncomfortable identifying as gay to the researcher. Thirteen of the 23 GSWs living with HIV agreed to talk to the researcher.

All the GSWs expressed concerns about people’s reactions when they admit to living with HIV. This is why the researcher was careful to first establish trust and to explain the purpose of the research study before approaching the subject. All participants were informed about their right to confidentiality and made aware that they could withdraw from the interview at any stage and had the right to refrain from answering any questions that they did not wish to answer.

Prior to participating in the interviews, the researcher confirmed again that the participants understood the research aims and protocols and consented to being interviewed. The participants provided written informed consent. The author used an interview guide to facilitate and focus the interview, but conversation was not restricted to the guide. All interviews were conducted in Putonghua, the mother tongue of both the participants and the researcher. Conversing in Putonghua helped build trust and enabled the participants to speak freely and colloquially about their situations. The GSWs appeared to be relaxed, comfortable, and willing to have a frank discussion about their lives after contracting HIV. Upon completion of the interview, each participant received 200 yuan (US$30) as an incentive for participation.

Data comprised recorded interviews, in situ note taking, and post-event field notes. Each interview was transcribed, translated into English, then uploaded into Nvivo 11.0. The subsequent analysis was driven primarily by a codebook. This was conceptualized through a re-reading (and coding) of the interview’s initial line of questions, followed by a separate coding of the responses obtained from the interviewees themselves. A thematic analysis was used to identify clusters of responses about specific topics. Quote excerpts and coding memos were developed according to themes. Representative and verbatim quotes were selected to illustrate key findings.

This article uses pseudonyms and slightly modified biographies to mask the participants’ identities. It helps the story to ‘flow’ better with names instead of markers or numbers. Some verbatim quotes may read a bit odd because some Chinese expressions do not smoothly translate to English. Effort has been made to maintain the spirit and intention of what was said wherever possible. The ethical approval of this research protocol came from the author’s Institutional Review Board (reference number: 3-9-202003-04).

## 3. Results

### 3.1. Felt Stigma and Living With HIV

The GSWs living with HIV were acutely aware of felt stigma. They explicitly criticized themselves, and even expressed hatred towards sex work, HIV, and homosexuality. Most of the stigma was expressed as shame and regret. In addition, due to the fear of felt stigma, most of the GSWs did not reveal their homosexuality work as a prostitute to parents and relatives; they typically used drugs to escape thinking about it.

The 25 GSWs described feelings of anxiety and mixed emotions about themselves which lasted at least half a year. Mafan says

I did not let my parents nor my partners know I have HIV. I hate myself… I hate HIV. I am afraid inside. One day, I could not tolerate my heart struggle, and disclosed to my parents that I am gay and a GSW. They were so calm as they kicked me out of our village. I take methamphetamine quite often I liaise with some drug dealers and smugglers in Yunnan… There’s a strong demand for drugs from communities like ours. Sometimes I don’t care enough to use condoms after taking drugs.

Since reaching adulthood, Linqu (25) was regularly pressured by his parents to get married and have children. After a positive experience disclosing his sexual orientation to his boyfriend’s parents, he felt the time was right for him to openly share with his parents. He and his boyfriend visited his parents and discovered that he was wrong,

At the very beginning my parents didn’t know what I meant. Finally, I just told them I am gay and held my boyfriend’s hands. Then they finally understood. My father was so angry, he was so tense, he quickly stood up and then collapsed to the floor. He stood up a second time and began yelling and pointing at us. He told my boyfriend to get out of our village immediately. My mother couldn’t stop crying. Later, my mother told me my father had a heart attack and was hospitalized. She begged me to get married and have a kid. I didn’t want to disappoint my mother, so I went and married a girl and finally we had a son. Since that time, I only visit home during the lunar Chinese New Year.

Shuo (22) also married a heterosexual woman after being regularly pressed by his mother. A powerful socio-cultural norm that was repeatedly brought up was the way the older parents continually urge their children to make them grandparents. He says,

My parents want me to have a normal life, get married, and have a kid. I failed them. I still think that failing to get married is a disgrace and sign of disrespect to my parents. Who is going to inherit my bloodline, particularly in rural China? I lose face before my relatives and family members during the Spring Festival. My neighbors even think my parents did something wrong since I don’t have kids. My mom even wants me to **“**rent**”** a bride to make her happy. She calls me twenty times per day to urge me to get married.

The severe consequences of ‘coming out’ explain why GSWs choose to keep their sexuality and occupation a secret from their family members, and why GSWs abuse drugs to cope with the harsh realities of life.

### 3.2. Enacted Stigma Against GSWs Living With HIV

Majian, a 36-year-old gay sex worker living with HIV for two years, reflects that much of the felt stigma comes from government officials:

I don’t have sympathy from my parents because I contracted HIV from my ex-partner. Most people blame us for our sexually risky behaviour… If they know I have HIV, they will stare and keep a distance from me. Therefore, I don’t dare to let them know. Some government officials consider me as “scum” (*feiwu* 廢物) and say we do not deserve to receive proper treatment. The department of health and the security department (e.g., police) think we are troublesome because we increase their workload.

Menrui (28) has been infected HIV for five years. He worked in a HIV NGO as a volunteer for two years. He told the author that he understands that every December 1, only the provincial CDC will publish the data reported by the Health Department. He says the government is playing statistical trickery, for example, saying only two people were infected and died of HIV even though many bodies were dumped in a landfill. Or they may attribute several deaths to tuberculosis or general pneumonia as a way to avoid widespread panic. Menrui (28) says,

The government did not really want to see us. They want us to be quiet, did not lodge a complaint though we don’t enjoy medical care in urban China. They did not like us as we make the city look bad and dirty. We are the liability and stumbling stone of China’s modernity! The discrimination against HIV comes from the government, medical doctors, and cadres. I can tell the government just wants to dump us into a landfill, ship it to the outlying island to let us die and be forgotten. The government doesn’t care about us.

Mao (28) has been living with HIV for two years. Mao said the government needs clear guiding principles on how to treat people with HIV. Currently, most of the money spent on NGOs goes towards HIV testing. However, how best to heal the HIV patients remains uncertain and lacks clear guidelines. Mao (28) says,

The hospital I visit in my hometown is highly decentralized and localized, and act depending upon their own needs to treat the HIV patients. I saw the hospital use a mattress to roll up dead bodies and send them to burn in the landfill. They did not even inform the patients’ family because HIV remains so sensitive in China. They reported the patients died of something like skin allergy or cancer rather than HIV. I know they do not treat HIV victims as human…

The Chinese authorities have treated HIV more as a medical and law enforcement issue than a social issue. There are good national legislations protecting the rights of the people living with the disease but such laws are not properly implemented locally. The government’s mentality is to divide the population into those who have HIV (e.g., sex workers, drug users and MSM) and those who do not and isolate them from each other in order to contain the disease. However, the virus inevitably spreads, most commonly through unprotected sex with close companions. GSWs are high-risk group and the government only considers their HIV infection rate in terms of how it might affect stability and harmony in Chinese society. Treatment, medication, and counselling services are almost undeveloped in urban China. The deeply ingrained traditions and values result in social avoidance, reflecting enacted stigma towards GSWs living with HIV. The GSWs often initially struggle with felt stigma but seem to accept their situation over time. Social forces declare that, since they do not embody the stereotypical ideal portrayed in state-sanctioned notions of masculinity, they must be pushed to the fringe of today’s China.

### 3.3. Enacted Stigma from the Medical Doctors

Although medical and allied health care professionals in China are required by law to treat patients who are living with HIV, enacted stigma permeates the health care system. The relationship between underprivileged GSWs and the Communist medical system’s use of necropolitical power to destine homosexual bodies to death—particularly those who are living with HIV—can now be better understood.

Jianhui, aged 31, contracted HIV and saw doctors regularly to get his PrEP and other medication. Jianhui (31) explained how humiliating the experience could be:

To get the PrEP, I have to repeatedly beg the CDC to prescribe the medicine to me. If I don’t beg the doctor, he won’t give it to me. The doctors show their true color by refusing to prescribe. They said I look fine. I am in good shape and there is no need to take medication. But it is obvious, my skin still has a rash and my lungs are bad. How else am I to get better?! Begging to the doctor was no fun.

According to Jianhui, during his 10 years of HIV treatment in China, he observed that hospitals often discourage health workers from using gloves and masks due to cost concerns. Occupational exposure to blood (and thus blood-borne viruses, BBV) is high, while compliance with standard precautions among health workers is low/suboptimal. There are few public reports about incidents, explaining why health workers in non-designated hospitals often refuse to accept HIV/AIDS patients.

Chen (24) is a GSW, but he repacked everything and moved back to his hometown to treat his HIV due to enacted stigma received from the doctors and officers. He says,

Discrimination from doctors and officers hurt me a lot. I think the data on deaths from HIV are adjusted in advance before they are sent to the central authorities. Some GSWs like me, we want revenge. We feel numb, desperate, and tired about the discrimination from the government and health system. Condom use is still not a habit for me even though I’m HIV positive. I really don’t care. Once, I had a male client who, on several occasions, put knock-out drops into water for me to drink, then during my prolonged unconsciousness, would have sex with me... I woke up discovering he was penetrating me. I did not even have the chance to let him know I am HIV positive. The client did not deliberately trick me as I knew he was going to do it. But I didn’t care. I still refused to let him know I am a HIV positive.

GSWs living with HIV face further risks in addition to discrimination from the health service. In China, GSWs must cope with multi-layered forms of enacted and felt stigma towards homosexuality and their work as gay prostitutes. The participants in this study grapple with chronic felt stigma as they continue hiding their sexual orientation from family members for fear of reprisal, persecution, and mental anguish. To cope, they typically resort to drug use, which in turn feeds into enacted stigma. Enacted stigma towards GSWs with HIV in China comes from several sectors, including government policies that promote traditional heterosexuality and police and security who monitor and periodically make arrests. Together, felt and enacted stigmas create necropolitics—the message to GSWs is that medical professionals and the wider society need to be protected from them. Since governments often justify a few deaths to protect the majority, the biopolitical control of the population is an inevitable part of a wider necropolitical project.

### 3.4. Rural Hukou Excluded from Urban Medical Care

The politics of health care in China places GSWs outside the medical care system. Having a rural *hukou* (the household registration system) makes the GSWs living with HIV marginalized urban citizens and “socially dead.” As a permanent identification number, the *hukou* can inadvertently be used to alienate the HIV-positive GSWs from their family home, or village. This is testament to the fact that GSWs are placed in a state of precarious ‘social death.’ The stigma attached to HIV is also felt as pressure from China’s medical services, which discriminate against rural GSWs. In the absence of state support or policies to alleviate their suffering and hardship, the abandonment of GSWs living with HIV will undoubtedly continue.

Yanquan (38) had been HIV positive for approximately one month and described the differences in treatment,

Some urban residents living with HIV can get the Triumeq (antiretroviral drug used in China) which contains three ingredients in a pill. A month prescription of Triumeq costs around $450 yuan (US$73). However, I received some inexpensive medicine from the rural hospital. It is neither efficient nor effective. It is just to keep my life going a little bit longer. Some locals can also get a drug ‘cocktail’ which contains multiple pills to combat HIV. I have nothing and try to use traditional Chinese medicine to ease symptoms.

Wingzhong (28) shared a story similar to that of Yanquan. The *hukou* of Wingzhong (28) is in rural China, at an isolated, secluded, small town. To get his medication, it takes an entire day of travel. He says,

Some middle-class gay communities receive better medication for HIV and find it is easy to go to the USA to heal. This is not my case. Each month, I have to take a 13-hour (round trip) high speed train to pick up my medication from my hometown. Each time, it takes at least 5 hours to consult with the doctor and then wait for the medicine. It is very inconvenient. But what else can I do? I don’t have an urban *hukou.*

Rui (29) experienced the same situation with his rural *hukou*. He says,

There is nothing I can do in the city as I don’t have an urban *hukou* and I am low-educated. I am not allowed to get HIV medicine from urban hospitals or clinics. Each time, it takes more than 10 hours on the high-speed railway because I live in a remote village in North China. I feel like the HIV medications in my hometown are insufficient. It seems the medicines don’t do anything to help. I feel like those hospitals in town provide better services and medication. I know some of my friends obtain higher medical allowance than in my hometown. I guess this is because of my rural *hukou*…

Sheng (34) was kicked out of a relatively comprehensive medical care system in urban China and sent him back to his hometown although he is a legal citizen in China. He is eligible to live and work in urban China but he is deprived of comprehensive medical care in urban China. He says,

It takes 17 hours to travel from my village to the nearest hospital to get the PEP. I was so sick I could not take the high-speed train to see the doctor. Therefore, I stayed in the city. I spent US$1500 ($9300 yuan) to buy the PEP from the drug stores. Those 30 days after taking PEP, waiting, were a dark side for me. I lost 30 pounds and refused to see anyone, putting myself under quarantine while I took the PEP. However, all my money and effort proved futile as the result was positive…So then I have two options. One is go back to my mediocre medical service in my hometown in rural China. The other option is spending all my money but get the better HIV treatment in the city. I am stressed, hopeless, and desperate. I am not optimistic about overcoming HIV!

The fear of discrimination is universal; it is found in conservative and collectivist countries such as China as well as Western democracies known for tolerant attitudes towards homosexuality. The data reported here confirm that there are deeply ingrained structures of *enacted* stigma which resulted in a reluctance to seek medical help from doctors and staff.

### 3.5. Marginalized Citizens under China’s Medical Care System

Health care professionals and nurses in dependency wards face a high risk of being personally infected by patients. Such was the case in 2003 when the SARS virus erupted in Hong Kong and Southeast Asia. In the wider international literature on stigma, Hafeez et al. [31] notes that, although the United States is tolerant towards non-conforming sexualities, health care providers nevertheless can stigmatize the LGBTQ community by a lack of awareness about the challenges faced by those who need treatment. Therefore, even inadvertent stigma from medical professionals can lead to mental and physical health disparities suffered by LGBTQ communities.

In China, there is a notable lack of awareness among police and medical officers regarding the causes of HIV infection. HIV patients must go to a hospital specifically designated by the government for that purpose. The HIV problem in China is compounded by depersonalization; infected people are discussed only as statistics, which makes it difficult to make urgent and specific changes. For example, shortages of medical supplies such as masks and gloves becomes debated as a general funding issue. However, without masks and gloves, doctors may refuse to touch infected patients. The government focus on aggregate numbers removes the human aspect, which is the focus of patients and medical workers.

Kang, 28, has experienced the problem first hand,

I know when I visit my doctor every month, he told me they are broke and did not receive enough funding from the government. There is a shortage of protective equipment. Therefore, my treatment is quite mediocre. The medication won’t heal AIDS immediately, but it won’t let me die either. The doctor laughed at me and said “We have to reuse goggles, surgical gowns, and even shoe covers. Sometimes, I even have to re-use my N95 surgery mask. The shortage of medical supplies prevents me from helping HIV patients. I am on the front line fighting the HIV epidemic”. My medical doctors yelled at me and said I was naïve to think doctors could help me. I can become the casualty because of government missteps and administrative failures. This puts me at risk of infection if I have an urgency to operate a surgery but most of the doctors refuse to help. What they can do is to prescribe some mediocre medications. The medications neither heal nor help.

Stigma from medical professionals is partly influenced, as Kang explained, by how the Chinese government treats HIV-positive individuals as well as gay and lesbian individuals. When people are treated as case numbers—mere digits—it reflects the non-humanness of the homosexual body. Referring to people in the aggregate provides the authorities a mechanism with which to exercise biopolitical control. Acknowledging this aspect of stigma, some HIV-positive GSWs agreed that medical professionals should look after themselves and their families first.

Maya (34) has lived with HIV for 5 years, and says he cannot afford the recommended ‘cocktail therapy’ combining zidovudine, lamivudine, and efavirenz to combat HIV because it costs approximately 200,000 yuan (US$29,411) a year,

Many mainland doctors are reluctant to go to rural areas. I only have rural *hukou* and cannot afford ‘cocktail therapy.’ The government now issues monthly allowances—about 50 yuan (US$7.5)—to the HIV patients in North China. It is not much but it’s better than nothing.

The stigma towards GSWs who are pathogenically diseased has been conditioned by the attitude and funding of the government—the attitude towards GSWs living with HIV frames them as deserving of stigma and being left to die. The intersection between the living dead homosexual body and the enacted stigma from the medical profession is the manifestation of necropolitical power. The body of the GSW, as a potential or an actual carrier of HIV, is already a body assigned death or what Mbembe [4] calls the ‘living dead’—GSW deaths will not constitute any form of moral or financial loss to society, rather this end is framed as a means to achieve the greater good. In this worldview, medical doctors caring for HIV patients are deemed more worthy of life.

## 4. Discussion

Necropolitical mechanisms are responsible for negatively impacting the palliative outcomes or imminent death of GSWs, homosexuals with HIV, or individuals who are potential HIV carriers. However, the data also confirm that even medical professionals can be unintended victims of the biopolitical control of GSWs. We should de-center the body of the pre-HIV homosexual body in the city as one that is marked for biopolitical exclusion before it is exposed to necropolitical brutality. Enacted stigma in urban China is inflicted upon these rural-to-urban migrants who are also homosexual. This discrimination subjects GSWs to biopolitical retaliation. In the words of Davies, Isakjee and Dhesi [7]: “the permanent wounding of individuals, rather than their direct and active killing, can be used as a means of control.” Thus, stigma towards GSWs living with HIV and denied health access is a cruel apparatus, where they are “kept alive but in a state of injury” [4].

These data show how stigma fuels biopolitical modes of governing homosexuals and peasants and is intrinsically linked to instances of leaving them to die. The findings also demonstrate how biopolitical measures can negatively affect collateral groups—not only homosexuals, but also at-risk heterosexual medical workers. Mbembe’s notion of necropolitics provides critical insight into contemporary conditions in China’s health care system, where all involved may be exposed to death. Homosexuality and other gender non-conforming forms of behavior expose GSWs to state-sanctioned stigma that supports biopolitical control and punishing them for being HIV positive. Although the sexual activity of GSWs is consensual, they are fully conscious of necropolitical power treating them as the living dead for failing to uphold ideals of masculine Chinese citizenship. As bodies constituting the ‘necessary other’ among heterosexual hegemony, lesbians, gays and transgenders in China are vanquished to the private and embodied domain.

## 5. Conclusions

Necropower of state-sanctioned biopolitical control is an effective tool for managing GSWs and those infected with HIV while securing the future of the state to control peasant male migrants who enjoy the benefits of urban China. The dynamics of the politics of health care in the Chinese system includes the voices of the medical doctors, GSWs living with HIV, and NGO directors. The politics of death assigned to the homosexual body are invariably indistinguishable from enacted stigma that is institutionalized in the medical system. To this end, the distinctions drawn by Scambler [22] reveal that enacted stigma is an important part of necropolitical power, which creates multiple death worlds through psychological trauma and anxiety through felt stigma. An expectation of discrimination from medical professionals and family members discourages homosexuals from accessing public health care services and heightens their susceptibility to death through other pathologies arising from chronic stress.

Further research should operationalize the necropolitics arising from stigma in the health services as well as in sociology and cultural studies of non-conforming gender and sexuality in China. Gendered notions of citizenship in China have produced a set of biopolitical conditions which insulate the population from perceived threats such as GSWs, MSM, lesbians, and transsexuals. As such, members of the LGBTQ community are subject to *social death* as non-human and non-living entities. Enacted and felt stigma function as biopolitical measures to protect the moral order and future of Chinese society. However, this stigma can backfire and create necropolitical conditions for medical doctors who must treat patients with HIV infections. Future research must address how the medical system is part of a much wider biopolitical and necropolitical environment, whereby effeminate boys are excluded from the Chinese Dream. The mechanisms of necropower confirm why it is important to reduce stigma in China’s health service and the wider society towards these excluded groups in urban China.

Non-stigmatization of vulnerable groups is a human-rights issue in China. The felt and enacted stigmas experienced by the rural GSWs makes them precarious citizens, marginalized as socially dead. The government position seems to be, since nobody is directly killed or murdered, no one can be held responsible. However, the experience of GSWs living with HIV reflects a social death, outsourced to penal colonies, an ‘extraordinary rendition’ which becomes ordinary, obfuscated by state bureaucracy, and covered up by one media spectacle after another [32]. International organizations can amplify the voice of the NGO community, urging public policy changes in China. However, it is difficult to attract international attention when public opinion is muzzled. Public health and infectious disease management require multi-dimensional policies and voices, instead of relying on official voices from the state apparatus. State-controlled health care in China reflects the government message that justice is less important than control, and control is the key to crack down on a contagious disease such as HIV. Our humanity demands that we keep raising awareness about the exclusion of HIV-infected GSWs from the medical health care system in China [33].

The culturally-ingrained practice of expressing enacted stigma towards homosexuality in China prevents MSW from ‘coming out.’ [34] They will continue to hide their orientation and occupation from their family members. For GSWs who are living with HIV and struggling with social necropolitics, non-governmental organizations (NGOs), state actors, and other social service providers can consider how to reduce the social stigma associated with homosexuality as well as address the cultural expectation for men to give their parents a grandchild.

China generally displays ambivalence towards the interconnected issues of (1) gender and sexual diversity; (2) the segregation between urban *hukou* and rural *hukou*; (3) heteronormativity and homophobia; (4) heteronormative national-cultural expectations of family, gender, and marriage; and (5) stigma towards the LGBTQ community. China’s rural LGBTQ community has expressed concern that they are treated as adjuncts of the country, with their rural background and sexuality conscripted into fulfilling the nation’s longstanding ideal to project social harmony [35]. For example, GSWs complain that they have been denied not only medical services, but also political, social, and economic opportunities [36]. The necropolitics of social death should open debates among policy makers—for example, how sovereignty and state power assign bodies without bio-value to necropolises, as examples where deaths do not matter and are of no concern to even the urban gentrified gay communities [37,38]. The notion of the necropolitics of social death helps socio-medical researchers understand how the loss of political rights when denied appropriate health care creates death worlds where individuals are left to die. Some GSWs even hide their HIV history and still have sex with their clients. Therefore, GSWs urgently need more counseling services, workshops, and education to avoid spreading HIV to their clients and sex partners [39].

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
