# Peer review of "Gay Sex Workers in China’s Medical Care System: The Queer Body with Necropolitics and Stigma"

_ijerph, 2020, doi:10.3390/ijerph17218188_

Round 1

Reviewer 1 Report

See detailed comments in PDF attached.

Specifically:

This is a very heavily theoretical paper - the terminology and theory needs clearer introduction and framing - most readers will not be familiar with them. The language used is highly theoretical and needs to be framed, explained, and simplified for the reader - especially for a public health audience, this will be very inaccessible. This brings me to question if this article is suited to a public health journal and audience - although some of the subject matter, and the effects of the stigmatisation and discrimination have an impact on the health of this population, this framing of the issue may be better suited to a more theoretical / social theory journal. 

It is unusual for literature to be cited in the results section. A results section should just be a presentation of findings from the present study. Literature should be cited in the discussion section.

Reviewer 2 Report

Although the author indicates that this is a 4-years study, I do not see the these 4 years of work on this topic when reading the results section (just some sentences and expressions from 28 gay sex workers from China which mainly describes the stigma they suffer.

Although the topic is of greater interest, honestly I do not see the contribution this paper makes to the field of sexuality and HIV.

Reviewer 3 Report

I think this is a very interesting and original paper with a genuinely sociological focus. It might not make easy reading for non-sociologists, but the articulation of concepts and theses is detailed and clear. I like the application of the ideas of necropolitics and stigma to the problems confronting gay men with HIV in China. I also like the way the author covers each of: (1) the everyday problems members of her sample experience in the lifeworld, (2) the views and procedures emanating from the medical profession, and (3) the health policy and social order issues of the Chinese state.  The result is a serious piece of research with important implications for present and future health policy and practice.

I have only one suggestion, which I would be disinclined to 'insist' on. To ease the path of readers of the journal unfamiliar with the concept of necropolitics (in particular), it might be helpful for a more straightforward explication at the outset. But, as I say, I think the article is ok for publication as it is. A good and significant piece of work.

Round 2

Reviewer 1 Report

This manuscript has been much improved. I have made some minor suggestions in the attached PDF. 

One overall comment, which is really my opinion, and not necessarily subject to whether this paper should be published or not. But I find this discourse of social death detracts from the utility and accessibility of this paper's findings - which for me can be framed much more cleanly and simply within discourses of intersecting stigmas. Discourses of marginality can still apply - but I find that overall the overly academic and theoretical concepts of necropolitics and social death serve to cloud the clarity of the findings and their application to inform public health policy.

Reviewer 2 Report

The authors have made the changes.

Some references do not appear, please revise.
